# Updated Food Composition Database for Cereal-Based Gluten Free Products in Spain: Is Reformulation Moving on?

**DOI:** 10.3390/nu12082369

**Published:** 2020-08-07

**Authors:** Violeta Fajardo, María Purificación González, María Martínez, María de Lourdes Samaniego-Vaesken, María Achón, Natalia Úbeda, Elena Alonso-Aperte

**Affiliations:** Departamento de Ciencias Farmacéuticas y de la Salud, Facultad de Farmacia, Universidad San Pablo-CEU, CEU Universities, Urbanización Montepríncipe, Alcorcón, 28925 Madrid, Spain; violeta.fajardomartin@ceu.es (V.F.); mar.martinez1.ce@ceindo.ceu.es (M.M.); l.samaniego@ceu.es (M.d.L.S.-V.); achontu@ceu.es (M.A.); nubeda@ceu.es (N.Ú.); eaperte@ceu.es (E.A.-A.)

**Keywords:** gluten-free products, celiac disease, gluten-free diet, gluten containing products, food composition database

## Abstract

We developed a comprehensive composition database of 629 cereal-based gluten free (GF) products available in Spain. Information on ingredients and nutritional composition was retrieved from food package labels. GF products were primarily composed of rice and/or corn flour, and 90% of them included added rice starch. The most common added fat was sunflower oil (present in one third of the products), followed by palm fat, olive oil, and cocoa. Only 24.5% of the products had the nutrition claim “no added sugar”. Fifty-six percent of the GF products had sucrose in their formulation. Xanthan gum was the most frequently employed fiber, appearing in 34.2% of the GF products, followed by other commonly used such as hydroxypropyl methylcellulose (23.1%), guar gum (19.7%), and vegetable gums (19.6%). Macronutrient analysis revealed that 25.4% of the products could be labeled as a source of fiber. Many of the considered GF food products showed very high contents of energy (33.5%), fats (28.5%), saturated fatty acids (30.0%), sugars (21.6%), and salt (28.3%). There is a timid reformulation in fat composition and salt reduction, but a lesser usage of alternative flours and pseudocereals.

## 1. Introduction

One percent of the general population in the Western world is affected by celiac disease (CD), one of the most common food intolerances in Europe [1]. CD is an autoimmune disorder with an aberrant response to gluten proteins with subsequent atrophy of intestinal villi, impaired intestinal absorption, and malnutrition. Extra-intestinal symptoms such as fatigue, iron deficiency, and neurological/psychological disorders (e.g., depression) may also be present. Long-term risks associated with CD, such as lymphoma, osteoporosis, and anemia, have also been reported [2].

A strict and lifelong adherence to a gluten free diet (GFD) is the first-line treatment and, currently, the only effective therapy for celiac patients and all other gluten related disorders, such as non-celiac gluten sensitivity or wheat allergy [3]. Gluten originates from a family of proteins found in wheat (gliadins and glutenins), rye (secalins), barley (hordeins), and oats (avenins), or in their hybridized strains (e.g., spelt or kamut) [4]. A GFD comprises only naturally gluten free (GF) food products (e.g., legumes, fruit and vegetables, unprocessed meat, fish, eggs, dairy products, and GF cereals, such as rice or corn) and/or substitutes of wheat-based foods, specially manufactured without gluten or having a gluten content lower than 20 ppm, as per European legislation [5]. For the traditional gluten-containing foods, such as bakery products, there is currently a wide variety of GF options available that use GF cereals (rice, corn, millet, and sorghum) and pseudocereals (quinoa, buckwheat, amaranth, and teff) as their base ingredients [6]. However, a GFD is difficult to follow because gluten is an ingredient widely used in the food industry, appearing in products that originally do not contain gluten such as meat, fish, and many other foodstuffs [7]. Hence, product labels and ingredient lists need to be carefully reviewed.

Consumer’s interest and demand has led to a significant increase in the production and sales of GF products. Global market data indicate that GF product sales are forecasted to increase by a compound annual growth rate of 7.6% between 2020 and 2027 [8]. In the last decade, Spain has been the leader country increasing its production of GF goods (18.8%), compared to Western Europe and the rest of the world (13.6% and 15.4%, respectively), and has become the third world producer of this type of products, after U.S.A. and Brazil. In 2019, the European region held the maximum market share in the GF products market [7]. Reasons for this growth are not only due to purchases by those with CD or those with a gluten sensitivity but are also propelled by changes in consumer attitudes towards health. Mainstream consumers are experimenting with their diets for health-related reasons, and “free-from” foods (such as GF foods) are part of that trend [9]. 

However, comprehensive nutritional composition data of GF products, mainly vitamin and mineral content, are still scarce or limited [10,11]. More importantly, access to such data is even more restricted, since there is a broad lack of micronutrient data in food composition tables, databases, and food labels. This statement warrants the need of providing new data on mineral and vitamins in GF food products, to complete food composition tables or databases, to cover regulatory purposes, and/or to assess population dietary intakes [12]. Composition data are useful to evaluate the adequacy of nutrient intake of celiac patients, on which the debate is still open, and are, therefore, strongly needed [13]. 

The GFD has demonstrated benefits in managing some gluten-related disorders, although nutritional imbalances have been reported. Although a GFD is associated with being healthier by some authors [9,14], epidemiological studies indicate nutritional imbalances in different celiac populations following a GFD, both in children [13,15] and in adults [16,17]. They refer to both macronutrients and micronutrients, including minerals. Overall, nutritional imbalances include high lipid, high protein, and low fiber intakes, and lack of adequacy to reference intakes of vitamin D, calcium, and magnesium [13,15]. Celiac patients may also be at risk of iron and folate deficiencies [17]. Some authors state that nutritional deficiencies in CD patients may be due to GF products, which are made with highly refined flours and high amounts of fat and sugar to achieve a texture resembling the typical and unique viscoelastic properties of wheat [18,19]. 

To provide better consumer information, the objective of this paper is to develop a nutritional food composition database including cereal-based GF products available in Spain. For this purpose, we estimated the nutritional composition of the products considering both the nutritional information and the ingredients reported on the product label, focusing on the critical components that define the nutritional quality of a GFD (added flours, starches, fats, sugars, and fiber).

## 2. Materials and Methods 

### 2.1. Design and Data Collection

The present study involved the compilation of cereal-based gluten-free (GF) products available in the Spanish market. Products and brands were gathered systematically from manufacturer websites and/or specialized retail stores and supermarkets with the highest market shares in Spain between September 2016 and March 2019. Retailers such as Carrefour, Hipercor, Mercadona, Alcampo, and Lidl, as well as smaller specialized stores, were visited. Information on ingredients and nutritional composition was retrieved from the food package labels. Major commercial and distribution brands were selected. All products included in the study showed one of the following claims on the package: the European Crossed Grain Trademark, the Spanish Federation of Coeliac Associations (FACE) crossed grain symbol, or the “gluten free” claim. 

### 2.2. Food Database Development 

The cereal-based GF food database was developed according to LanguaL™ Thesaurus EuroFIR [20]. In total, 629 cereal-based GF food items were categorized into four groups, nine subgroups, and thirteen subgroup categories in the developed database, in consonance with the LanguaL™ classification. The four groups were beverages, milk, milk product, or milk substitutes, grain or grain products, and miscellaneous food products.

The grain or grain products group comprised six subgroups: bread and similar, breakfast cereals, cereal or cereal-like milling products and derivatives, fine bakery ware, pasta and similar products, and savory cereal dishes. Beverages (non-milk) included alcoholic beverages. The milk, milk product, or milk substitutes group included data from frozen dairy desserts. Finally, miscellaneous food products involved prepared food products. 

Each cereal-based GF food item was assigned to one of the following subgroups and categories: beer or beer-like beverages, frozen dairy desserts, bread products, leavened breads, unleavened breads, crisp breads, and rusks, breakfast cereals and cereal bars, cereal or cereal-like milling products and derivatives, biscuits, sweets, and semi-sweets, pancakes or waffles, pastries and cakes, pasta and similar products, pasta dishes, pies, unsweetened, or pizzas, savory cereal dishes, and savory snacks (see Table 1). Bread products included breadcrumbs. Leavened breads included rolls, buns, breads baked in pans and French type. Cereal or cereal-like milling products and derivatives included flour and flour preparations for baking products.

### 2.3. Food Composition in Terms of Ingredients

The ingredient list used in the formulation of the GF products was analyzed. Four groups of critical ingredients were considered: starchy ingredients (i.e., flour or starch), fats (oils and fats), sugars (i.e., dextrose), and added fibers (i.e., xanthan gum). Ingredients were chosen according to their impact on the nutritional profile of GF products. In each case, the top ten most frequently used ingredients were considered.

### 2.4. Nutritional Information Study

The nutritional composition of each product item included in the database is given in terms of quantity of energy and nutrients per 100 g of product as sold. Energy expressed in kcal, macronutrients (fats (g), saturated fatty acids (g), carbohydrates (g), sugars (g), protein (g)), fiber (g), and salt (g) were the data on nutrient composition reported on the label of each product. Micronutrients, vitamin, and mineral contents were not declared on the label in hardly any GF products. 

### 2.5. Statistical Analysis

Descriptive data on ingredients are expressed as frequency (number of products including a specific ingredient and percentage based on the total products within the category or the subgroup). Data on nutrient composition are expressed as average and standard deviation. 

## 3. Results

GF products available in Spanish markets were systematically compiled between September 2016 and March 2019. In total, 629 cereal-based GF products were studied, and each food item was assigned to one of nine subgroups, based on LanguaL™ Thesaurus 2017 (Figure 1). 

The main group was fine bakery ware (*n* = 229; 36.4%), followed by bread and similar products (*n* = 152; 24.2%) and pasta and similar products (*n* = 88; 13.9%). Minor categories were alcoholic beverages (*n* = 14; 2.2%) and frozen dairy desserts (*n* = 6; 0.9%). The targeted GF products belonged to more than 70 different commercial brands. Among the top five manufacturers of GF products (Schär, Santiveri, Airos, Adpan, and Proceli), four of them are Spanish companies. However, it should be noted that the leading producers were mostly different across different GF product categories. We developed an initial GF product database in 2016, including 271 cereal-based GF food products. Up to 10% of the foodstuffs found in the present update are no longer available, and 24.5% have been reformulated. 

The database is available for research purposes on demand.

### 3.1. Food Composition in Terms of Ingredients

Cereal-based GF products were primarily composed of rice and/or corn flour, and almost 90% of them included added rice starch (Table 1 and Table 2). Less than 10% of the GF products were formulated with other kinds of flours, such as buckwheat, soy and other legumes, brown rice, millet, or quinoa. Oatmeal, sorghum, amaranth, teff, guar, chia, chestnut, flax, or potato flours were very rarely present in the ingredient list. Corn starch was present in the formulation of 60% of the products, and other commonly added were rice and potato starches. Tapioca starch, modified starch, and potato maltodextrin were also found in some products. Barley malt was the main cereal used for beer-like beverages.

Considering bread products, wheat is the main flour used in Spain, but it must be substituted by rice flour, followed by corn flour, and frequently added with rice or corn starches when baking GF breads. Corn flour is more frequently used than rice flour when preparing GF breakfast cereals, pasta, and savory snacks. Soya flour was most frequently used in fine bakery ware, with 26% of biscuits, sweets, and semi-sweets containing this type of flour. Added starches were found in all products except for breakfast cereals and beer-like beverages.

One hundred thirty different combinations of added fats were found in the recorded GF food products. The most added fat was sunflower oil, which was present in almost one third of the products, followed by palm fat, olive oil, and cocoa, all used similarly in around 13% of the products. Other animal fats (butter, cream, or lard), margarines, rapeseed oil, and coconut oil were more seldom used (Table 3). Therefore, unsaturated fats were predominant in most of the GF foodstuffs considered in the database. However, palm fat was the main fat added to biscuits, sweets, and semi-sweets, classified in the fine bakery ware subgroup, in addition to other saturated fats such as cocoa and animal fats. When focusing on frozen dairy desserts, it was observed that only saturated fats were present (animal fats, coconut oil, palm oil, and cocoa). Both results are in accordance with the high amounts of fat and saturated fats found, respectively, in the subgroups in the macronutrient analysis (see Section 3.2). The three most used margarines were made by: palm, coconut, and sunflower; palm, coconut, and rapeseed; and coconut with sunflower. Bread and similar products and fine bakery ware were the subgroups in which margarines were more frequently used. Taken together, palm fat and margarines made up with palm oil were present in 22.8% of the GF products. Finally, no added fats or oils were used as ingredients in pasta and similar products, according to the labeling.

Only 154 GF foodstuffs (24.5%) had the nutrition claim “no added sugar” on the label [21], being pasta and similar products the most representative subgroup of this fact (Table 4). Among the added sugars, we found that 55.8 % of the GF products had sucrose in their formulation. Other sugars and sweeteners less employed were, in order of frequency, glucose, fructose, dextrose, and lactose. Other rich sugar ingredients such us non-refined or cane sugar, rice syrup, beetroot sugar syrup, and honey were also present in the GF products. Leavened breads, biscuits, sweets, semi-sweets, pastries, and cakes were the subgroups were GF products more frequently contained added sugars, with almost 100% of the products containing sucrose, dextrose or glucose, and fructose. Very few breakfast cereals and fine bakery ware (<3%) included no calorie sweeteners. 

Table 5 shows the type of fibers used in the formulation of GF products. Xanthan gum was the most frequently employed fiber, appearing in 34.2% of the GF products, followed by other commonly used such as hydroxypropyl methyl cellulose (23.1%), guar gum (19.7%), vegetable gums (psyllium, bamboo, chicory, potato, rice, pea, corn, etc.) (19.6%), and sodium carboxymethyl cellulose (6.4%). Fibers less frequently found in GF products were citrus fiber, carrageenan, pectin, cellulose, locust bean gum, and apple fiber (appearing in 1.6 to 2.7% of the products). The least fiber enriched products were breakfast cereals and pasta and similar products, whereas the most frequently supplemented were bread and similar products, fine bakery ware, and savory cereal dishes. Macronutrient analysis (Table 6) revealed that 25.4% of the products could be labeled as a source of fiber (>3 g/100 g), mostly breads, breakfast cereals, milling products, and fine bakery ware.

### 3.2. Nutritional Information

The highest amount of energy, total fats and sugars was found in the fine bakery ware subgroup (426.1 ± 77.7 kcal/100 g, 20.5 ± 6.8 g/100 g and 22.2 ± 9.2 g/100 g, respectively). The highest content of saturated fats was found in frozen dairy desserts (8.8 ± 4.2 g/100 g); proteins in savory cereal dishes (7.7 ± 2.0 g/100 g); carbohydrates in pasta and similar products (76.5 ± 5.9 g/100 g); and fiber and salt in bread and similar products (5.2 ± 2.2 g/100 g, 1.5 ± 0.5 g/100 g, respectively). Average salt content in all products was 0.6 ± 0.4 g/100 g. Highest contents were found in bread and similar products, savory cereal dishes, and prepared food products (Table 6). 

Many of the considered GF food products showed very high contents of: energy (33.5%), defined as >400 kcal/100 g; fats (28.4%), defined as >17.5 g/100 g; saturated fatty acids (30.0%), defined as >5 g/100 g; sugars (21.6%), defined as >22.5 g/100 g; and salt (28.3%), defined as >500 mg of sodium or the equivalent value for salt /100 g. On the other hand, 25.4% could be labeled as a source of fiber (>3 g/100 g) [21,22]. 

## 4. Discussion

This cross-sectional study of 629 cereal-based GF products represents the largest comparative nutrient analysis of packaged Spanish GF food products and their ingredients, up to date. Most of the considered GF food products (~ 30%) showed very high contents of energy, fats, saturated fatty acids, sugars, and salt. In contrast, 25.4% could be labeled as a source of fiber [21,22]. Compared to other recent studies, our food composition database showed similar or slightly lower nutrient values than others [23,24,25,26,27].

It is important to mention that there is not an unequivocal nutritional profile for GF food products worldwide. Differences from country to country, from brand to brand, and among food categories have been asserted. Furthermore, differences could be attributable to different methodology (product selection) between studies. Nutritional values of each food item included in the present database were calculated as average of all the single similar foods from each brand included in each category. Therefore, nutritional composition variability for each food item due to its ingredient formulation has been considered. 

Regarding ingredients, we found that the main fat component of GF products was sunflower oil, followed by palm fat, olive oil, and cocoa fat. This result differs to that shown by Calvo-Lerma et al. [25] in a similar study conducted in Spain, in which they found that GF products were largely composed of palm oil. Our database was developed up to March 2019 and the data collection in the study by Calvo-Lerma et al. [25] was conducted between March and October 2017. In this sense, this could indicate recent food reformulation to improve the nutritional quality of fat, thus providing a healthier food choice for celiac patients. In fact, recent research on palm fat and oil has brought up intriguing health issues, due to the presence of toxic contaminants generated in the processing of palm oil and other vegetable oils. This has promoted an update on the tolerable daily intakes (TDI) for toxic contaminants (2- and 3-monochloropropanediols and glycidyl esters [28,29]), but has also posed some misunderstandings in mass communication [28]. Consumers have been aware of these issues through the media, demanding and purchasing only palm oil free biscuits, especially for children. Processed foods are constantly changing as manufacturers try to protect or increase market share and profits and respond to policy changes dictated by a combination of government policies and consumer pressure, e.g., reduction of sugar, salt, saturated, and trans fatty acids [30]. General awareness of the role of diet on health is boosting the rate of changes in composition and foods consumed in many countries.

In the case of breads, we found that fat was commonly added to leavened breads, being it sunflower oil in more than 50% of the cases, followed by olive oil and a margarine made with palm, coconut, and sunflower oils. Consequent nutritional composition renders a considerably high amount of fat (5.7 %), most of it unsaturated (61%). Our data are slightly lower than those given by Calvo Lerma et al. [25] for total fat in a study of 619 GF products conducted in 2017. Miranda et al. [10] also studied Spanish GF commercialized breads in a study of 206 GF products, undertaken between 2012 and 2013, and stated that these contained less protein and double the fat content, (being this fat mainly SFA), in contrast to their gluten containing counterparts. Again, producers may be reducing fat in leavened breads, although data show that there is a large variability in fat composition when comparing different brands.

Fat content in pastries, cakes, pancakes, or waffles was 30% saturated and the most commonly added fat was palm fat. Results are in agreement with other studies [10,25]. 

GF products are made with high amounts of fat and sugar to achieve a texture resembling the typical and unique wheat viscoelastic properties [18,19]. Fat ingredients are indeed useful in bakery products for the stabilization of gas bubbles and the reduction of kneading resistance and swelling of starch granules [31]. Moreover, emulsifiers can be used to increase dough stiffness, improve bread structure, and decrease the speed of staling. In pasta products, emulsifiers act as lubricants in the extrusion process and provide firmer consistency and a less sticky surface, as they control starch swelling and leaching phenomena during cooking [32]. Other components such as sugars (sucrose, glucose, and fructose syrup), starch (corn starch, rice flour, and corn flour), and fibers (xanthan gum, hydroxypropyl methyl cellulose, and guar gum) were also present in GF products in our database.

It is interesting to point out that almost all breads contained added sugars (98% of leavened breads and 79% of unleavened breads, crisps, and rusks). Sugar is not a common ingredient in bread. In Spain, normal bread is solely composed of flour (usually wheat), salt, baker’s yeast, and water [33], although sugar may be added to special breads. Sugar addition in bread is normally a matter of concern since bread is not usually associated with sugar consumption in the population. Due to the sugar addition, simple carbohydrate composition raised to values around 5.2 g per 100 g. This amount is similar to that described by Miranda et al. [10] in 2012–2013, and somewhat smaller than that described by Calvo Lerma et al. [25] in 2017. Nonetheless, both authors state that there is no significant difference in sugar content when comparing GF breads with their gluten containing counterparts. Therefore, gluten-containing flour may be contributing to sugar composition on a higher amount as compared to GF flour. To prove this idea, sugar content in GF pasta, without added sugars, was quite low (below 1%), and other authors have demonstrated that GF pasta contains a significantly lower amount of sugars as compared to gluten containing pasta [10,25]. 

Cereal-based GF products were primarily composed of rice and/or corn flour. Less than 10% of the GF products were formulated with other kind of flours, such as buckwheat, soy and other legumes, brown rice, millet, or quinoa. Therefore, the type of flour used results in a high glycemic index, because of the high content (70–80%) of amylopectin and related glucose polymers. The use of pseudocereals is still small, although several authors present them as good gluten free alternatives. According to Jastrebova and Jägerstad [34], the best way to develop nutritious healthy GF products with high content of proteins, fibers, micronutrients, and antioxidants, is natural fortification by using nutritious ingredients such as whole grain flours of GF cereals/pseudocereals, protein-rich flours of soy, lupin, chick-pea, chestnut, and different seeds, as well as bioprocessing, such as germination or fermentation with yeast and/or sourdough. Other authors suggest the use of pseudocereals such as amaranth, quinoa, or buckwheat because of their content in thiamine, vitamin E, or carotenoids [35] or the nutritional quality of their protein, fat, fiber, and minerals [36].

In our study, soy, legume, and quinoa flours were present in 8.4%, 7.3% and 4.1%, respectively, of the analyzed products. Breakfast cereals were the group with the most frequent inclusion of alternative cereals such as teff, oatmeal, sorghum, and other flours coming from chia, amaranth, or flaxseeds. Again, manufacturers seem to be timidly introducing the use of nutritious pseudocereal and legume flours in the formulation of GF products.

Xanthan gum and hydroxypropyl methylcellulose are the most popular hydrocolloids that are used in GF products. They display thickening properties through the binding of water and, as a result, the viscosity of the gluten-free dough is enhanced and gas is better retained, improving loaf volume and structure [14].

Differences between GF products and their equivalents with gluten have also been described in other studies. The most recent surveys on the nutritional quality of GF food products currently available on the market, and recently reviewed by Melini and Melini [18], show key inadequacies—a low protein content and a high fat and salt content—compared to their equivalent gluten-containing products. However, an interesting trend towards some improvements has emerged. More adequate levels of fiber and sugar than in the past have been reported in the surveys of the last two years, although the composition in terms of fiber and sugars is highly variable between the different product categories. Further studies are nevertheless required to investigate the micronutrient content of GF food products, since very few reported data exist. Kulai and Rashid [37] and Jamieson et al. [26] informed of a significant lower iron and folate content in GF products compared to gluten containing food. Potassium content was also significantly lower in GF food products [27]. Furthermore, only 5% of GF breads were fortified with all four mandatory fortification nutrients (calcium, iron, nicotinic acid or nicotinamide, and thiamin), and 28% of GF breads were fortified with calcium and iron only in UK [24]. Fortified GF products represent only 10% of GF staple foods in Europe, because the use of starches (with low levels of many essential micronutrients) as main ingredient in many GF foods makes it difficult to implement common fortification with single micronutrients [34]. This lack of fortification may increase the risk of micronutrient deficiency in coeliac sufferers according to these authors. GF choices could account in unanticipated health disorders for CD patients based on the limited labeling description and narrow range of nutritionally balanced products and brands currently available [25].

Average salt content in all products was 0.6 g/100 g. As compared to other studies [10], we found much lower amounts of salt in pasta, cereal milling products, and fine bakery ware. Other studies [25] do not show data on salt content since salt content in nutritional information labels was only introduced as compulsory from December 2016 [38]. As with fat quality, salt reduction could be another of the reformulation targets that are being assessed in GF products.

Several population studies, in different countries, have investigated the nutritional status of CD patients adhering to a GFD. In published studies, CD patients consumed more fats (especially saturated), protein, and simple carbohydrates (sugars) but less fiber and micronutrients, such as iron, calcium, and vitamin D than recommended [11,17,39,40], and also compared to healthy subjects [16,41,42]. A research group from our laboratory recently showed no relevant differences in the general nutrient quality of the diet of children and adolescents following a GFD, as compared to matched controls, in contrast to previous studies, with the exception of polyunsaturated fatty acids, folate, and calcium intakes. These were significantly lower in coeliac as compared to non-coeliac children and adolescents, as well as low when compared to the recommended intakes for these nutrients [13]. Adequacy of vitamin D intake to recommendations was dramatically low, for both coeliac and non-coeliac children and adolescents; however, only coeliac girls presented a significantly lower level of plasmatic vitamin D (below reference values, <30 ng/mL), as compared to non-coeliac controls, although without clinical repercussion in bone mass density [13]. Therefore, we consider that vitamin D fortification in GF products could be a strategy of great importance to minimize adverse health effects associated to vitamin D deficiency. 

## 5. Conclusions

In conclusion, the present study represents an attempt to build a systematic composition database of GF products based on the ingredients listed on the label and the nutritional information provided by the manufacturer. This type of study is a priority, since CD patients include this type of products in their diets, and studies assessing CD patient’s diets need to use updated data on GF product composition. Moreover, since nutritional deficiencies have been described for CD patients and it has been shown that nutritional quality of GF products is lower, updated quality assessment of available products is needed for further improvement in GF product development. We describe 629 cereal-based GF products available in the Spanish market, in terms of ingredients and strategic nutrients. However, information on micronutrient composition is a still pending question.

### 5.1. Strengths

Studies that evaluate the formulations of commercially available GF products are scarce. In our study, we included an important number (629) of products and we describe them using a standardized classification (LanguaL™ Thesaurus EuroFIR), in order to make them comparable to other studies. Brands have also been recorded. Our data are likely to be just as accurate as most data reported for any kind of food product present on the market.

### 5.2. Limitations

Some limitations related to our data should be considered. First, since the nutritional composition data of GF products has been estimated, it cannot substitute a direct analysis. Direct chemical analysis is the gold standard to estimate the nutrient composition of food. Additionally, nutrient data shown on food labels provided by the food industry may be based on estimations from the ingredients rather than direct chemical analysis of the food products. Finally, another limitation is related to the lack of information on micronutrient content (minerals and vitamins) in GF products.

## Figures and Tables

**Figure 1 nutrients-12-02369-f001:**
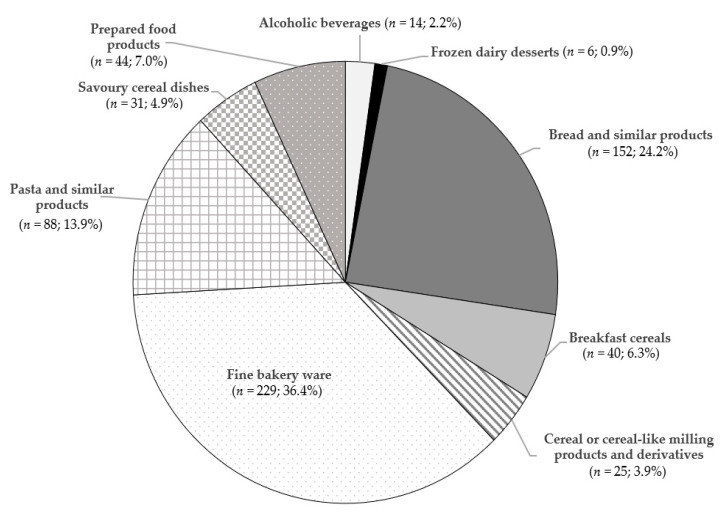
Cereal-based gluten free (GF) products included in the database (*n* = 629).

**Table 1 nutrients-12-02369-t001:** Types of flour used in the formulation of gluten free (GF) products.

Food Groups	Food Subgroups (*n*)	Rice *n* (%)	Corn * *n* (%)	Brown Rice *n* (%)	Millet *n* (%)	Buckwheat *n* (%)	Soy *n* (%)	Other Legumes ** *n* (%)	Quinoa *n* (%)	Barley Malt *n* (%)	Other Flours *** *n* (%)
Beverages (non-milk)	Alcoholic beverages (14)	2 (14.3)	5 (35.7)	0	1 (7.1)	0	0	0	0	10 (71.4)	0
	Beer or beer-like beverages (14)	2 (14.3)	5 (35.7)	0	1 (7.1)	0	0	0	0	10 (71.4)	0
Milk, milk product, or milk substitutes	Frozen dairy desserts (6)	4 (66.6)	2 (33.3)	0	0	0	1 (16.7)	4 (66.7)	0	0	0
Grain or grain products	Bread and similar (152)	88 (57.9)	36 (23.7)	13 (8.6)	22 (14.5)	22 (14.5)	3 (2.0)	15 (9.9)	10 (6.6)	0	18 (11.8)
	Bread products (11)	4 (36.4)	5 (45.5)	0	1 (9.1)	1 (9.1)	1 (9.1)	0	0	0	0
	Leavened breads (89)	50 (56.2)	12 (13.5)	7 (7.9)	19 (2.1)	11 (12.4)	2 (2.2)	7 (7.9)	8 (9.0)	0	10 (11.2)
	Unleavened breads, crisp breads, and rusks (52)	34 (65.4)	19 (36.5)	6 (11.5)	2 (3.8)	10 (19.2)	0	8 (15.4)	2 (3.8)	0	8 (15.4)
	Breakfast cereals (40)	18 (45.0)	29 (72.5)	1 (15.5)	2 (5.0)	4 (10.0)	1	0	4 (10.0)	0	10 (25.0)
	Breakfast cereals (35)	15 (42.9)	26 (74.3)	1 (2.9)	0	3 (8.6)	1 (2.9)	0	4 (11.4)	0	8 (22.9)
	Cereal bars (5)	3 (60.0)	3 (60.0)	0	2 (40.0)	1 (20.0)	0	0	0	0	2 (40.0)
	Cereal or cereal-like milling products and derivatives (25)	11 (44.0)	11 (44.0)	1 (4.0)	0	2 (8.0)	0	3 (12.0)	0	0	2 (8.0)
	Fine bakery ware (229)	99 (43.2)	80 (34.9)	7 (3.1)	5 (2.2)	7 (3.1)	42 (18.3)	20 (8.7)	0	0	9 3.9)
	Biscuits, sweets, and semi-sweets (96)	57 (59.4)	64 (66.7)	1 (1.0)	5 (5.2)	3 (3.1)	25 (26.0)	8 (8.3)	0	0	10 (10.4)
	Pancakes or waffles (6)	0	0	0	0	1 (16.7)	0	1 (16.7)	0	0	0
	Pastries and cakes (127)	42 (33.1)	16 (94.1)	6 (4.7)	0	3 (2.4)	17 (13.4)	11 (8.7)	0	0	0
	Pasta and similar products (88)	54 (61.4)	81 (92.0)	9 (10.2)	7 (7.9)	4 (4.5)	0	1 (1.1)	7 (8.0)	0	2 (2.3)
	Savory cereal dishes (31)	19 (61.3)	19 (61.3)	1 (3.2)	2 (6.5)	8 (25.8)	2 (6.5)	2 (6.5)	0	0	1 (3.2)
	Pasta dishes (13)	8 (61.5)	8 (61.5)	0	1 (7.7)	0	2 (15.4)	2 (15.4)	0	0	0
	Pies, unsweetened, or pizzas (18)	11 (61.1)	11 (61.1)	1 (5.6)	1 (5.6)	8 (44.4)	0	0	0	0	1 (5.6)
Miscellaneous food products	Prepared food products (44)	10 (22.7)	20 (45.5)	9 (20.5)	0	0	4 (9.1)	1 (2.3)	5 (11.4)	0	0
	Savory cereal dishes (16)	6 (37.5)	4 (25.0)	0	0	0	0	0	0	0	0
	Savory snacks (28)	4 (14.3)	16 (57.1)	9 (32.1)	0	0	4 (14.3)	1 (3.6)	5 (17.9)	0	0
Total (629)		305 (48.5)	283 (45.0)	41 (6.5)	39 (6.2)	47 (7.5)	53 (8.4)	46 (7.3)	26 (4.1)	10 (1.6)	43 (6.8)

Results are expressed as frequency (*n*, number of products including a specific ingredient) and (percentage based on the total products within the category or the subgroup). * Includes corn flour, corn grits, or corn. ** Other legumes include chickpea, pea, carob, lentil, lupine, or fava bean. *** Other flours include oatmeal, sorghum, amaranth, teff, guar, chia, chestnut, flax, or potato.

**Table 2 nutrients-12-02369-t002:** Types of starch used in the formulation of gluten free (GF) products.

Food Groups	Food Subgroups (*n*)	Corn *n* (%)	Rice *n* (%)	Potato *n* (%)	Tapioca *n* (%)	Modified *n* (%)	Modified Corn *n* (%)	Modified Tapioca *n* (%)	Modified Potato *n* (%)	Potato Maltodextrin *n* (%)	Wheat Starch without Gluten *n* (%)
Beverages (non-milk)	Alcoholic beverages (14)	0	0	0	0	0	0	0	0	0	0
	Beer or beer-like beverages (14)	0	0	0	0	0	0	0	0	0	0
Milk, milk product, or milk substitutes	Frozen dairy desserts (6)	3 (50.0)	1 (16.7)	1 (16.7)	0	0	0	0	0	0	0
Grain or grain products	Bread and similar (152)	134 (88.2)	33 (21.7)	17 (11.2)	18 (11.8)	1 (0.7)	2 (1.3)	0	0		1 (0.7)
	Bread products (11)	10 (90.9)	0	0	1 (9.1)	0	0	0	0	0	0
	Leavened breads (89)	85 (95.5)	28 (31.5)	4 (4.5)	15 (16.9)	1 (1.1)	0	0	0	0	0
	Unleavened breads, crisp breads, and rusks (52)	39 (75.0)	5 (9.6)	13 (25.0)	2 (3.8)	0	2 (3.8)	0	0	0	1 (1.9)
	Breakfast cereals (40)	0	0	0	0	0	0	0	0	0	0
	Breakfast cereals (35)	0	0	0	0	0	0	0	0	0	0
	Cereal bars (5)	0	0	0	0	0	0	0	0	0	0
	Cereal or cereal-like milling products and derivatives (25)	17 (68.0)	1 (4.0)	4 (16.0)	3 (12.0)	1 (4.0)	0	0	0	0	0
	Fine bakery ware (229)	182 (79.5)	58 (25.3)	74 (32.3)	14 (6.1)	5 (2.2)	16 (7.0)	11 (4.8)	1 (0.4)	6 (2.6)	4 (1.7)
	Biscuits, sweets, and semi-sweets (96)	64 (66.7)	26 (27.1)	46 (47.9)	2 (2.1)	0	12 (12.5)	11 (11.5)	1 (1.0)	6 (6.3)	0
	Pancakes or waffles (6)	6 (100)	0	0	0	0	0	0	0	0	1 (16.7)
	Pastries and cakes (127)	112 (88.2)	32 (25.2)	28 (22.0)	12 (9.4)	5 (3.9)	4 (3.1)	0	0	0	3 (2.4)
	Pasta and similar products (88)	2 (2.3)	0	4 (4.5)	0	0	0	0	0	0	0
	Savory cereal dishes (31)	19 (61.3)	10 (32.3)	7 (22.6)	0	6 (19.4)	2 (6.5)	0	0	0	7 (22.6)
	Pasta dishes (13)	12 (92.3)	3 (23.1)	4 (30.8)	0	3 (23.1)	1 (7.7)	0	0	0	0
	Pies, unsweetened, or pizzas (18)	7 (38.9)	7 (38.9)	3 (16.7)	0	3 (16.7)	1 (5.6)	0	0	0	7 (38.9)
Miscellaneous food products	Prepared food products (44)	20 (45.5)	2 (4.5)	11 (2.3)	2 (4.5)	2 (4.5)	4 (9.1)	4 (9.1)	0	0	0
	Savory cereal dishes (16)	12 (75.0)	1 (6.3)	9 (56.3)	2 (12.5)	2 (33.3)	2 (12.5)	0	0	0	0
	Savory snacks (28)	8 (28.6)	1 (3.6)	2 (7.1)	0	0	2 (7.1)	4 (14.3)	0	0	0
Total (629)		377 (59.9)	105 (16.7)	118 (18.7)	37 (5.9)	15 (2.4)	24 (3.8)	15 (2.4)	1 (0.2)	6 (1.0)	12 (1.9)

Results are expressed as frequency (*n*, number of products including a specific ingredient) and (percentage based on the total products within the category or the subgroup).

**Table 3 nutrients-12-02369-t003:** Fat ingredients used in the formulation of gluten free (GF) products.

Food Groups	Food Subgroups (*n*)	Sunflower Oil *n* (%)	Palm Fat * *n* (%)	Olive Oil *n* (%)	Cocoa *n* (%)	Animal Fats (Butter, Cream or Lard) *n* (%)	Margarine 1 (Palm, Coconut and Sunflower) *n* (%)	Rapeseed Oil *n* (%)	Coconut Oil *n* (%)	Margarine 2 (Palm, Coconut and Rapeseed) *n* (%)	Margarine 3 (Coconut and Sunflower) *n* (%)
Beverages (non-milk)	Alcoholic beverages (14)	0	0	0	0	0	0	0	0	0	0
	Beer or beer-like beverages (14)	0	0	0	0	0	0	0	0	0	0
Milk, milk product, or milk substitutes	Frozen dairy desserts (6)	0	2 (33.3)	0	2 (33.3)	4 (66.7)	0	0	3 (50)	0	0
Grain or grain products	Bread and similar (152)	57 (37.5)	7 (4.6)	23 (15.1)	1 (0.7)	0	18 (11.8)	9 (5.9)	1 (0.7)	0	12 (7.9)
	Bread products (11)	0	1 (9.1)	2 (18.2)	0	0	0	0	0	0	0
	Leavened breads (89)	45 (50.6)	1 (1.1)	12 (13.5)	1 (1.1)	0	14 (15.7)	3 (3.4)	0	0	10 (11.2)
	Unleavened breads, crisp breads, and rusks (52)	12 (23.1)	5 (9.6)	9 (17.3)	0	0	4 (7.7)	6 (11.5)	1 (1.9)	0	2 (3.8)
	Breakfast cereals (40)	9 (22.5)	6 (15.0)	0	5 (12.5)	1 (2.5)	0	1 (2.5)	1 (2.5)	0	0
	Breakfast cereals (35)	6 (17.1)	4 (11.4)	0	3 (8.6)	0	0	1 (2.9)	1 (2.9)	0	0
	Cereal bars (5)	3 (60.0)	2 (40.0)	0	2 (40.0)	1 (20.0)	0	0	0	0	0
	Cereal or cereal-like milling products and derivatives (25)	0	0	0	0	0	0	0	0	0	0
	Fine bakery ware (229)	98 (42.8)	61 (26.6)	35 (15.3)	68 (29.7)	41 (17.9)	11 (4.8)	8 (3.5)	16 (7.0)	18 (7.9)	8 (3.5)
	Biscuits, sweets, and semi-sweets (96)	22 (22.9)	38 (39.6)	21 (21.9)	24 (25.0)	10 (10.4)	3 (3.1)	4 (4.2)	11 (11.4)	14 (14.6)	0
	Pancakes or waffles (6)	1 (16.7)	0	0	0	4 (66.7)	0	0	0	0	0
	Pastries and cakes (127)	75 (59.1)	23 (18.1)	14 (11.0)	44 (34.6)	27 (21.2)	8 (6.3)	4 (3.1)	5 (3.9)	4 (3.1)	8 (6.3)
	Pasta and similar products (88)	0	0	0	0	0	0	0	0	0	0
	Savory cereal dishes (31)	19 (61.3)	4 (12.9)	13 (41.9)	0	5 (16.1)	2 (6.5)	4 (12.9)	1 (3.2)	0	0
	Pasta dishes (13)	8 (61.5)	3 (23.1)	6 (46.2)	0	5 (38.5)	0	0	1 (7.7)	0	0
	Pies, unsweetened, or pizzas (18)	11 (61.1)	1 (5.6)	7 (38.9)	0	0	2 (11.1)	4 (2.2)	0	0	0
Miscellaneous food products	Prepared food products (44)	17 (38.6)	4 (9.1)	13 (29.5)	7 (15.9)	6 (13.6)	6 (13.6)	1 (2.3)	0	4 (9.1)	0
	Savory cereal dishes (16)	10 (62.5)	2 (12.5)	1 (6.3)	0	6 (37.5)	6 (37.5)	1 (6.3)	0	0	0
	Savory snacks (28)	7 (25.0)	2 (7.1)	12 (42.9)	7 (25.0)	0	0	0	0	4 (14.3)	0
Total (629)		200 (31.8)	84 (13.4)	84 (13.4)	83 (13.2)	57 (9.1)	37 (5.9)	23 (3.7)	22 (3.5)	22 (3.5)	20 (3.2)

Results are expressed as frequency (*n*, number of products including a specific ingredient) and (percentage based on the total products within the category or the subgroup). * Palm fat includes palm kernel or palm stearin. Other fats not included are milk, egg (liquid or powder), or cheese.

**Table 4 nutrients-12-02369-t004:** Sugar addition and types of sugars and sweeteners used in the formulation of gluten free (GF) products.

Food Groups	Food Subgroups (*n*)	No Added Sugars Declared *n* (%)	Added Sugar Presence *n* (%)	Sucrose *n* (%)	Dextrose *n* (%)	Other Sugars * *n* (%)	Glucose and Fructose Syrup *n* (%)	Non-Refined or Cane Sugar *n* (%)	Rice Syrup *n* (%)	Beetroot Sugar Syrup *n* (%)	Honey *n* (%)	Lactose *n* (%)	No-Calorie Sweeteners *n* (%)
Beverages (non-milk)	Alcoholic beverages (14)	11 (79.0)	3 (21.0)	1 (7.0)	0	2 (14.0)	0	0	0	0	0	0	0
	Beer or beer-like beverages (14)	11 (79.0)	3 (21.0)	1 (7.0)	0	2 (14.0)	0	0	0	0	0	0	0
Milk, milk product, or milk substitutes	Frozen dairy desserts (6)	0	6 (100.0)	6 (100.0)	1 (16.0)	0	5 (83.0)	0	0	0	0	0	0
Grain or grain products	Bread and similar (152)	15 (10.0)	137 (90.0)	71 (47.0)	45 (30.0)	3 (2.0)	25 (16.4)	9 (6.0)	23 (15.0)	2 (1.3)	12 (8.0)	0	0
	Bread products (11)	2 (18.0)	9 (82.0)	6 (54.5)	1 (9.0)	0	2 (18.0)	1 (9.0)	0	0	0	0	0
	Leavened breads (89)	2 (2.0)	87 (98.0)	41 (46.0)	28 (31.0)	2 (2.0)	16 (18.0)	7 (8.0)	20 (22.4)	2 (2.0)	12 (13.0)	0	0
	Unleavened breads, crisp breads, and rusks (52)	11 (21.0)	41 (79.0)	24 (46.0)	16 (31.0)	1 (2.0)	7 (13.4)	1 (2.0)	3 (6.0)	0	0	0	0
	Breakfast cereals (40)	8 (20.0)	32 (80.0)	29 (72.5)	3 (7.5)	1 (2.5)	9 (22.5)	4 (10.0)	0	0	3 (7.5)	0	1 (2.5)
	Breakfast cereals (35)	8 (23.0)	27 (77.0)	24 (68.5)	2 (6.0)	0	5 (14.0)	4 (11.4)	0	0	1 (3.0)	0	1 (3.0)
	Cereal bars (5)	0	5 (100.0)	5 (100.0)	1 (20.0)	1 (20.0)	4 (80.0)	0	0	0	2 (40.0)	0	0
	Cereal or cereal-like milling products and derivatives (25)	11 (44.0)	14 (56.0)	6 (24.0)	7 (28.0)	0	1 (4.0)	3 (12.0)	0	0	0	0	0
	Fine bakery ware (229)	4 (2.0)	225 (98.0)	195 (85.0)	45 (20.0)	7 (3.0)	99 (43.0)	32 (14.0)	6 (3.0)	9 (4.0)	3 (1.3)	4 (2.0)	4 (2.0)
	Biscuits, sweets, and semi-sweets (96)	1 (1.0)	95 (99.0)	74 (77.0)	6 (6.2)	4 (4.1)	34 (35.4)	26 (27.0)	4 (4.1)	9 (9.3)	3 (3.1)	3 (3.1)	3 (3.1)
	Pancakes or waffles (6)	1 (17.0)	5 (83.0)	2 (33.3)	4 (66.6)	0	0	0	0	0	0	0	0
	Pastries and cakes (127)	2 (2.0)	125 (98.0)	119 (94.0)	35 (28.0)	3 (2.3)	65 (51.0)	6 (5.0)	2 (2.0)	0	0	1 (0.7)	1 (0.7)
	Pasta and similar products (88)	87 (99.0)	1 (1.0)	0	0	0	0	1 (1.0)	0	0	0	0	0
	Savory cereal dishes (31)	5 (16.0)	26 (84.0)	10 (32.2)	0	2 (6.4)	7 (22.5)	0	0	0	1 (3.2)	0	0
	Pasta dishes (13)	4 (30.0)	9 (70.0)	6 (46.1)	0	2 (15.3)	5 (38.4)	0	0	0	0	0	0
	Pies, unsweetened, or pizzas (18)	1 (5.5)	17 (94.4)	17 (94.4)	10 (55.5)	0	2 (11.1)	0	0	0	1 (5.5)	0	0
Miscellaneous food products	Prepared food products (44)	13 (30.0)	31 (70.0)	20 (45.4)	4 (9.0)	10 (23)	9 (20.4)	2 (4.5)	2 (4.5)	0	0	2 (4.5)	0
	Savory cereal dishes (16)	1 (6.0)	15 (94.0)	10 (62.5)	3 (19.0)	5 (31.2)	6 (37.5)	0	0	0	0	2 (12.5)	0
	Savory snacks (28)	12 (43.0)	16 (57.0)	10 (36.0)	1 (3.5)	5 (18.0)	3 (11.0)	2 (7.1)	2 (7.1)	0	0	0	0
	Total (629)	154 (24.5)	475 (75.5)	351 (55.8)	115 (18.2)	25 (4.0)	155 (24.6)	51 (8.1)	31 (5.0)	11 (2.0)	19 (3.0)	6 (1.0)	5 (0.7)

Results are expressed as frequency (*n*, number of products including a specific ingredient) and (percentage based on the total products within the category or the subgroup). * Other sugars include isomaltose, fructose, glucose, agave syrup, golden syrup, maltodextrin, or high maltose corn syrup.

**Table 5 nutrients-12-02369-t005:** Types of fibers used in the formulation of gluten free (GF) products.

Food Groups	Food Subgroups (*n*)	Xanthan Gum *n* (%)	Hydroxypropyl Methyl Cellulose *n* (%)	Guar Gum *n* (%)	Vegetable Gums * *n* (%)	Sodium Carboxymethyl Cellulose *n* (%)	Citrus Fiber *n* (%)	Carrageenan *n* (%)	Pectin ** *n* (%)	Cellulose *n* (%)	Locust Bean Gum *n* (%)	Apple Fiber *n* (%)
Beverages (non-milk)	Alcoholic beverages (14)	0	0	0	0	0	0	0	0	0	0	0
	Beer or beer-like beverages (14)	0	0	0	0	0	0	0	0	0	0	0
Milk, milk product, or milk substitutes	Frozen dairy desserts (6)	0	0	5 (83.3)	0	0	0	1 (16.6)	0	0	2 (33.3)	0
Grain or grain products	Bread and similar (152)	77 (50.7)	91 (59.9)	31 (20.4)	75 (49.3)	15 (9.9)	4 (2.6)	0	4 (2.6)	1 (0.7)	0	5 (3.3)
	Bread products (11)	6 (54.5)	4 (36.4)	3 (27.3)	0	0	0	0	0	0	0	0
	Leavened breads (89)	26 (50.0)	25 (48.1)	15 (16.9)	20 (38.5)	1 (1.9)	0	0	0	0	0	0
	Unleavened breads, crisp breads, and rusks (52)	5 (9.6)	9 (17.3)	12 (23.1)	0	0	0	6 (11.5)	1 (1.9)	2 (3.8)	0	4 (7.7)
	Breakfast cereals (40)	0	0	0	2 (5.0)	0	0	0	0	0	0	0
	Breakfast cereals (35)	0	0	0	2 (5.7)	0	0	0	0	0	0	0
	Cereal bars (5)	0	0	0	0	0	0	0	0	0	0	0
	Cereal or cereal-like milling products and derivatives (25)	4 (16.0)	4 (16.0)	5 (20.0)	1 (4.0)	6 (24.0)	0	2 (8.0)	0	0	0	1 (4.0)
	Fine bakery ware (229)	118 (51.5)	34 (14.8)	51 (22.3)	30 (13.1)	18 (7.9)	12 (5.2)	10 (4.4)	4 (1.7)	7 (3.1)	9 (3.9)	1 (0.4)
	Biscuits, sweets, and semi-sweets (96)	23 (24.0)	1 (1.0)	17 (17.7)	7 (7.3)	0	10 (10.4)	0	0	0	1 (1.0)	0
	Pancakes or waffles (6)	5 (83.3)	0	1 (16.7)	0	1 (16.7)	0	0	0	0	0	0
	Pastries and cakes (127)	90 (70.9)	33 (26.0)	33 (26.0)	23 (18.1)	17 (13.4)	2 (1.6)	10 (7.9)	4 (3.1)	7 (5.5)	8 (6.3)	1 (0.8)
	Pasta and similar products (88)	0	0	1 (1.1)	0	0	0	0	0	0		0
	Savory cereal dishes (31)	8 (25.8)	15 (48.8)	21 (67.7)	13 (41.9)	0	0	1 (3.2)	5 (16.1)	5 (16.1)	0	2 (6.5)
	Pasta dishes (13)	3 (23.1)	0	8 (61.5)	6 (46.2)	0	0	0	0	0	0	2 (15.4)
	Pies, unsweetened, or pizzas (18)	5 (27.8)	15 (83.3)	13 (72.2)	7 (38.9)	0	0	1 (5.6)	5 (27.8)	5 (27.8)	0	0
Miscellaneous food products	Prepared food products (44)	8 (18.2)	1 (2.3)	10 (22.7)	2 (4.5)	1 (2.3)	1 (2.3)	0	0	0	1 (2.3)	1 (2.3)
	Savory cereal dishes (16)	8 (50.0)	1 (6.3)	8 (50.0)	2 (12.5)	0	0	0	0	0	1 (6.3)	1 (6.3)
	Savory snacks (28)	0	0	2 (7.1)	0	1 (3.6)	1 (3.6)	0	0	0	0	0
Total (629)		215 (34.2)	145 (23.1)	124 (19.7)	123 (19.6)	40 (6.4)	17 (2.7)	14 (2.2)	13 (2.1)	13 (2.1)	12 (1.9)	10 (1.6)

Results are expressed as frequency (*n*, number of products including a specific ingredient) and (percentage based on the total products within the category or the subgroup). * Vegetable gums include psyllium, bamboo, chicory, potato, rice, pea, or corn. ** Extract from apple or other fruits.

**Table 6 nutrients-12-02369-t006:** Energy and nutrient composition per 100 g of gluten free (GF) products, based on the nutritional information on the labels.

Food Groups	Food Subgroups (*n*)	Energy (kcal)	Fats (Total) (g)	SFA (g)	Carbohydrates (g)	Sugars (g)	Protein (g)	Fiber (g)	Salt (g)
Beverages (non-milk)	Alcoholic beverages (14)	42.4 ± 12.5	0	0	4.2 ± 1.49	1.8 ± 1.5	0.3 ± 0.2	ND	0
	Beer or beer-like beverages (14)	42.4 ± 12.5	0	0	4.2 ± 1.49	1.8 ± 1.5	0.3 ± 0.2	ND	0
Milk, milk product, or milk substitutes	Frozen dairy desserts (6)	291.8 ± 115.6	12.5 ± 4.7	8.8 ± 4.2	33.1 ± 10.4	21.4 ± 3.7	3.3 ± 1.1	1.7 ± 0.7	0.1 ± 0.0
Grain or grain products	Bread and similar (152)	318.3 ± 69.4	6.8 ± 5.8	2.5 ± 3.1	58.7 ± 15.5	4.3 ± 3.2	3.1 ± 2.1	5.2 ± 2.2	1.5 ± 0.5
	Bread products (11)	369.0 ± 28.8	3.3 ± 3.6	1.1 ± 1.6	78.5 ± 7.1	2.8 ± 2.7	4.1 ± 2.1	3.1 ± 2.0	1.5 ± 0.5
	Leavened breads (89)	283.9 ± 50.7	5.7 ± 3.3	2.2 ± 2.6	52.4 ± 11.7	5.2 ± 3.4	2.7 ± 1.6	5.8 ± 1.9	1.4 ± 0.4
	Unleavened breads, crisp breads, and rusks (52)	366.5 ± 68.3	9.3 ± 8.2	3.2 ± 3.9	65.2 ± 16.6	3.0 ± 2.2	3.6 ± 2.6	4.5 ± 2.4	1.7 ± 0.6
	Breakfast cereals (40)	381.3 ± 35.9	5.4 ± 4.9	1.7 ± 1.8	73.0 ± 12.8	17.0 ± 11.4	7.5 ± 2.6	5.2 ± 3.4	0.6 ± 0.6
	Breakfast cereals (35)	385.0 ± 26.4	4.5 ± 4.5	1.3 ± 1.4	75.6 ± 9.2	15.4 ± 10.8	7.9 ± 2.5	5.2 ± 3.4	0.6 ± 0.6
	Cereal bars (5)	355.0 ± 75.3	12.2 ± 1.6	4.5 ± 1.8	54.9 ± 20.2	27.6 ± 10.2	5.1 ± 1.7	4.9 ± 3.9	0.2 ± 0.2
	Cereal or cereal-like milling products and derivatives (25)	337.1 ± 39.7	2.1 ± 2.5	0.7 ± 1.3	73.0 ± 13.1	8.1 ± 10.8	3.9 ± 2.6	4.5 ± 3.4	0.6 ± 0.7
	Fine bakery ware (229)	426.1 ± 77.6	20.5 ± 6.8	8.1 ± 5.4	55.3 ± 12.1	22.2 ± 9.2	4.1 ± 1.6	3.2 ± 2.1	0.7 ± 0.5
	Biscuits, sweets, and semi-sweets (96)	471.5 ± 41.8	19.9 ± 5.9	9.4 ± 5.5	67.5 ± 6.4	25.5 ± 8.3	4.4 ± 1.4	3.5 ± 2.3	0.6 ± 0.5
	Pancakes or waffles (6)	237.9 ± 122.2	9.3 ± 8.7	3.4 ± 4.2	36.5 ± 11.1	6.5 ± 10.9	1.7 ± 1.5	ND	0.8 ± 0.1
	Pastries and cakes (127)	400.6 ± 71.4	21.4 ± 6.8	7.3 ± 5.1	47.0 ± 8.1	20.5 ± 8.6	4.0 ± 1.7	3.0 ± 1.7	0.8 ± 0.6
	Pasta and similar products (88)	347.3 ± 25.5	1.3 ± 0.7	0.2 ±0.3	76.5 ± 5.9	1.0 ± 0.5	6.4 ± 1.2	2.2 ± 1.7	0.04 ± 0.1
	Savory cereal dishes (31)	224.5 ± 55.5	7.9 ± 2.9	3.5 ± 1.6	29.7 ± 15.2	2.4 ± 1.8	7.7 ± 2.0	2.7 ± 1.7	1.1 ± 0.5
	Pasta dishes (13)	216.0 ± 82.1	6.6 ± 3.4	2.9 ± 1.8	31.5 ± 23.1	2.9 ± 2.6	6.8 ± 2.0	2.3 ± 1.9	0.9 ± 0.6
	Pies, unsweetened, or pizzas (18)	230.6 ± 24.3	8.8 ± 2.0	4.0 ± 1.2	28.4 ± 5.3	2.1 ± 0.8	8.4 ± 1.8	3.0 ± 1.4	1.3 ± 0.4
Miscellaneous food products	Prepared food products (44)	351.1 ± 132.8	9.9 ± 7.5	4.1 ± 4.5	58.7 ± 25.0	6.5 ± 10.3	5.6 ± 2.2	2.4 ± 1.4	1.2 ± 0.7
	Savory cereal dishes (16)	196.9 ± 87.0	6.2 ± 4.9	2.1 ± 2.2	30.1 ± 18.1	1.9 ± 1.8	4.7 ± 2.1	1.4 ± 0.9	1.1 ± 0.5
	Savory snacks (28)	439.1 ± 42.1	12.0 ± 7.9	5.2 ± 5.1	75.1 ± 7.3	9.0 ± 12.1	6.1 ± 2.1	2.8 ± 1.3	1.2 ± 0.8
Total (629)		302.2 ± 62.7	7.4 ± 4.0	3.3 ± 2.5	51.4 ± 12.5	9.4 ± 5.8	4.7 ± 1.7	3.4 ± 2.1	0.6± 0.4

Data are expressed as average ± standard deviation. SFA, saturated fatty acids.

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
