# Peer review of "Updated Food Composition Database for Cereal-Based Gluten Free Products in Spain: Is Reformulation Moving on?"

_nutrients, 2020, doi:10.3390/nu12082369_

Round 1

Reviewer 1 Report

This is an informative paper, clearly written and scientifically sound. It has some elements of novelty such as the number of studied GF products (629) and the classification based on LanguaL™Thesaurus.

However, the manuscript requires some minor corrections before it is ready for publication. The list of suggested changes follows.

In the Title, it is suggested to add “ in Spain” after “products” because this is a database of products consumed in Spain.

In the abstract, in the last sentence it is stated that there is a reformulation: compared to what? Maybe it should be specified the term of reference

In the keywords the word “gluten” alone is not entirely correct: maybe it is better to use “Gluten containing products”

Line 38 better said “gluten originates from a family of…..”

Line 62 Why are you saying that access to such data is even more restricted? An explanation is necessary

Line 83 After “Spain” please add a list of names starting with “such as…..”

Line 91 After the first phrase, add a second one with the list of the four groups i.e. “The four groups were……”

Line 104 Add at the end “(see Table 1)”

Lines 123-125 This first phrase should be better positioned in 2.1 Design and data collection

Line 262 Show instead of shows

Line 272 Baking? Maybe you mean “cooking”: I understand we are talking of pasta here

Line 326 “for” instead of “in”

Line 338 and 346. Why are you saying “We”? I don’t see a perfect correspondence between the authors of this manuscript and those listed for reference [12] so it should be better said “ A research group from our laboratory” or a similar phrase

Line 348 Move “systematic” before “composition”

Line 365 Amongst the Limitations the following phrase should be added: “Another limitation is related to the lack of information on micronutrients (minerals and vitamins) content of GF products.

Reviewer 2 Report

July 30, 2020

Journal: Nutrients

Manuscript ID: nutrients-890465

Title: Updated food composition database for cereal-based gluten free products. Is reformulation moving on? 

Opinion

In this study, the authors have developed a comprehensive composition database of a huge number of (n= 629) cereal-based gluten-free products available in Spain by collecting information on ingredients and nutritional composition given on food package labels. The macronutrient analysis performed by the authors revealed that 25.4% of the products could be labeled as a source of fiber. Many of the considered gluten-free food products showed very high contents of energy (33.5%), fats (28.5%), saturated fatty acids (30.0%), sugars (21.6%), and salt (28.3%). The study is well structured and nicely written the data given in the tables is appropriate. I have only a few minor comments.

Comments:

Comment#1(Line no. 177): The added sugar definition is probably not as per the standard of the authorities (WHO, FAO, or USDA). The authors should provide appropriate_reference_for_the_definition.

Comment#2 (Line no.319): ''Potassium content was overall significantly lower in GF food products''. I am confident with this line. As far I know, the most important potassium sources in the diet are vegetables and fruits, legumes, and nuts. The cereals are not the main potassium source in the diet. If I am right, the potassium content of Gluten-free cereal would be lower.

Comment#3: Line no 48-49, and 69-70, need a reference.
